# FOXP3 Isoforms Expression in Cervical Cancer: Evidence about the Cancer-Related Properties of FOXP3Δ2Δ7 in Keratinocytes

**DOI:** 10.3390/cancers15020347

**Published:** 2023-01-05

**Authors:** Natalia Garcia-Becerra, Marco Ulises Aguila-Estrada, Luis Arturo Palafox-Mariscal, Georgina Hernandez-Flores, Adriana Aguilar-Lemarroy, Luis Felipe Jave-Suarez

**Affiliations:** 1Programa de Doctorado en Ciencias Biomédicas, Centro Universitario de Ciencias de la Salud, Universidad de Guadalajara, Guadalajara 44340, Mexico; 2División de Inmunología, Centro de Investigación Biomédica de Occidente (CIBO), Instituto Mexicano del Seguro Social (IMSS), Guadalajara 44340, Mexico

**Keywords:** FOXP3, FOXP3Δ2Δ7, cervical cancer, HPV, RNAseq, GSEA

## Abstract

**Simple Summary:**

FOXP3 is a critical transcription factor that works as a master regulator of the lymphoid lineage. The expression of FOXP3 was also observed in tumor cells; in cervical cancer, FOXP3 increases as the tumor progresses. However, the biological role of FOXP3 in cervical pathology is not well understood. In addition, FOXP3 has isoforms that could have different biological properties. In this work, the expression of FOXP3 and its isoforms were evaluated. It was found that the isoform FOXP3Δ2Δ7 is expressed in the cervical cancer-derived cell line SiHa. The transduction of this isoform in nontumorigenic keratinocytes induces proliferation, cell division, and migration. RNAseq analysis indicated that the FOXP3Δ2Δ7 isoform induces the expression of different protooncogenes and modulates essential pathways related to the immune response and the tumorigenic process.

**Abstract:**

Cervical cancer (CC) is the fourth most common type of cancer among women; the main predisposing factor is persistent infection by high-risk human papillomavirus (hr-HPV), mainly the 16 or 18 genotypes. Both hr-HPVs are known to manipulate the cellular machinery and the immune system to favor cell transformation. FOXP3, a critical transcription factor involved in the biology of regulatory T cells, has been detected as highly expressed in the tumor cells of CC patients. However, its biological role in CC, particularly in the keratinocytes, remained unclarified. Therefore, this work aimed to uncover the effect of FOXP3 on the biology of the tumoral cells. First, public databases were analyzed to identify the *FOXP3* expression levels and the transcribed isoforms in CC and normal tissue samples. The study’s findings demonstrated an increased expression of FOXP3 in HPV16+ CC samples. Additionally, the *FOXP3Δ2* variant was detected as the most frequent splicing isoform in tumoral cells, with a high differential expression level in metastatic samples. However, the analysis of *FOXP3* expression in different CC cell lines, HPV+ and HPV-, suggests no relationship between the presence of HPV and *FOXP3* expression. Since the variant *FOXP3Δ2Δ7* was found highly expressed in the HPV16+ SiHa cell line, a model with constitutive expression of *FOXP3Δ2Δ7* was established to evaluate its role in proliferation, migration, and cell division. Finally, RNAseq was performed to identify differentially expressed genes and enriched pathways modulated by *FOXP3Δ2Δ7*. The exogenous expression of *FOXP3Δ2Δ7* promotes cell division, proliferation, and migration. The transcriptomic analyses highlight the upregulation of multiple genes with protumor activities. Moreover, immunological and oncogenic pathways were detected as highly enriched. These data support the hypothesis that *FOXP3Δ2Δ7* in epithelial cells induces cancer-related hallmarks and provides information about the molecular events triggered by this isoform, which could be important for developing CC.

## 1. Introduction

Cervical cancer (CC) is one of the most preventable and treatable types of cancer due to the extended precancerous phase that lasts decades, yet it is still recognized as a health burden in low- and middle-income countries where it is ranked as the fourth most common type of cancer [1,2] and the second leading cause of cancer death, both in women and men [3]. The most frequent histopathologic subtype of CC is squamous cell carcinoma (SCC), which accounts for nearly 75–85% of the cases, and it is derived from the transformation of squamous cells at the external portion of the cervix, termed ectocervix, and the adenocarcinoma (ADC), with approximately 10–25% of CC cases, which is a consequence of columnar epithelial cells transformation at the endocervix, the internal portion of the organ [4,5,6]; there is also a third subtype named as adenosquamous carcinoma (ADSC), which accounts for merely 2–3% and is originated by a merge of both squamous and glandular cells [5].

The main risk factor for CC is persistent infection by high-risk human papillomavirus (hr-HPV), such as the 16 and 18 genotypes, which both are associated with approximately 70% of cases worldwide [7,8]; HPV16 is frequently associated with SCC, while HPV18 is associated with ADC [9]. Hr-HPVs stimulate cell transformation over decades through the constant expression of their oncoproteins to activate immunological and carcinogenic pathways to favor tumorigenesis and modulate local immunity [10,11,12,13]. 

FOXP3, forkhead box p3, is a member of the transcription factor family known as forkhead/winged helix; its locus is in the short arm of the X chromosome, which transcribes for an mRNA of 11 protein-coding exons. Interestingly, by alternative splicing, five isoforms could arise from the FOXP3 mRNA; these are *FOXP3-FL*, *FOXP3Δ2*, *FOXP3Δ7*, *FOXP3Δ2Δ3*, and *FOXP3Δ2Δ7* [14,15]. FOXP3 is recognized as a master regulator responsible for immune tolerance through the modulation of genetic and functional programs in regulatory T cells (Treg) [16]. The increase in Tregs in cancer is associated with a worse prognosis due to their immunosuppressive functions. The suppressive properties of Tregs rely on stable FOXP3 expression [16,17,18]. The two major isoforms of FOXP3 reported in Tregs are *FOXP3-FL* and *FOXP3Δ2*, which are expressed in approximately equal amounts; the main functional difference between both isoforms is derived from the inability of *FOXP3Δ2* to inhibit RORα and RORγt functions [19,20,21]. A recently published study suggested that the expression of FOXP3 is closely related to the occurrence and growth of cervical cancer [22]. The expression of FOXP3 correlates with the prognosis of cervical cancer, and it is significantly higher in cancer than in cervical intraepithelial neoplasia or chronic cervicitis [22]. However, it remains unanswered whether there is a correlation between Tregs FOXP3+ and hr-HPV infection [23]. Nowadays, it is known that the FOXP3 expression is not exclusive of lymphoid lineage as it has been detected in epithelial cells from the breast, lung, prostate [24], and retinal tissue [25], as well as in cancer cells [26,27]. 

The role of FOXP3 in tumor cells is, however, controversial. It has been observed in breast and prostate cancer cells that FOXP3 expression is associated with antitumoral roles [28,29,30,31]. Furthermore, in ovarian cancer cells, inhibitory properties in proliferation, migration, and invasion were observed when FOXP3 was upregulated [32]. Conversely, the FOXP3 increase has also been linked to protumor functions in pancreatic, colorectal, gastric, bladder, thyroid, cervical, and nonsmall cell lung cancer [33,34,35,36,37,38]. In CC, the FOXP3 expression increased as the lesion progressed [39]. In addition, the knockdown of FOXP3 diminishes the growth of the SiHa cells [40]. However, whether the FOXP3 expression is regulated by HPV, which isoform is expressed in normal and tumoral keratinocytes, and the implications of FOXP3 expression in nontumorigenic epithelial cells are questions that remain unanswered. Therefore, this work aimed to evaluate the role of FOXP3 in the biology of keratinocytes and its contribution to the development of CC. 

## 2. Materials and Methods

### 2.1. Cell Culture

SiHa, CaSki, HeLa, SW756, C33A, HaCaT, and Lenti-X 293T cell lines were cultivated in Dulbecco’s Modified Eagle Medium (DMEM) with D-glucose (4.5 g/L) (No. Cat. 10-013-CV, Sigma-Aldrich, St. Louis, MI, USA), L-glutamine (584 mg/L), penicillin (100 U/mL), streptomycin (100 μg/mL), sodium pyruvate (110 mg/L), and 10% fetal bovine serum (FBS). Cells were maintained at 37 °C and a 5% CO_2_ atmosphere in an incubator (C170UL-120V-R, Binder, Tuttlingen, Germany). 

### 2.2. FOXP3Δ2Δ7 Open Reading Frame Cloning

The open reading frame (ORF) of *FOXP3Δ2Δ7* was amplified through conventional PCR with DreamTaq Green PCR Master Mix (Cat. No. K1081, Thermo Scientific, Waltham, MA, USA) according to the manufacturers’ instructions; cDNA from SiHa cells was used as a template. Primer sequences were designed using the Primer-BLAST tool from NCBI [41] (Table 1). The PCR product (1347 pb) was resolved by electrophoresis in 0.8% agarose gel, cut, and purified with Zymoclean Gel DNA Recovery Kit (Cat. No. D4001/D4002, ZYMO RESEARCH Corporation, Irvine, CA, USA) according to their instructions. The amplicon was ligated into the pGEM-T Easy vector (Cat. No. A137A, Promega, Madison, WI, USA) with T4 DNA Ligase (Cat. No. M180A, Promega, Madison, WI, USA) overnight at 4 °C. The ligation reaction was transformed into competent TOP10 bacteria by thermal shock and selected by the white/blue method. The constructed plasmid was named pGEM-FOXP3Δ2Δ7; from this, plasmid preparations were performed using the QIAprep Spin Miniprep Kit (Cat. No. 27106, QIAGEN Inc., Valencia, CA, USA). 

Subsequently, the *FOXP3Δ2Δ7* ORF was isolated from the pGEM-FOXP3Δ2Δ7 vector by restriction with EcoRI (No. Cat. IVGN0116, Invitrogen, Waltham, MA, USA) and subcloned into the lentiviral vector pLVX-Puro (Cat. No. 632164, Clontech Laboratories Inc., Mountain View, CA, USA) that was previously linearized with EcoRI, and dephosphorylated with Antarctic Alkaline Phosphatase (Cat. No. IVGN2204, Invitrogen, Waltham, MA, USA). The resulting plasmid pLVX- FOXP3Δ2Δ7 was used to sequence the FOXP3Δ2Δ7 ORF using the Big Dye v3.1 Terminator Cycle Sequencing kit (Cat. No. 4337455, Applied Biosystems, Thermo Fisher Scientific Inc., Waltham, MA, USA), and the ABI PRISM 310 Genetic Analyzer (Applied Biosystems). The sequences were aligned using the UGENE software (v.1.21.1) [42]. The NCBI sequences NM_014009.3 and NM_001114377.1 were used as references.

Finally, the plasmid pLVX-FOXP3Δ2Δ7 was used to obtain lentiviral particles.

### 2.3. Production of Lentiviral Particles

Lenti-X 293T cell line (Cat. No. 632180, Clontech Laboratories Inc., Mountain View, CA, USA) was used as viral producer cells; 4 × 10^6^ cells were seeded in p100 plates and cultured for 24 h. Plasmids-lipofectamine 2000 complexes were prepared as follows, both lipofectamine 2000 reagent (Cat. No. 11668019, Invitrogen, Waltham, MA, USA) and plasmids pLVX-FOXP3Δ2Δ7, pCMVR8.74 (Addgene 22036), and pMD2.G (Addgene 12259) were separately diluted in DMEM-free FBS medium. The plasmids pCMVR8.74 and pMD2.G were used to produce the necessary packaging proteins for lentiviral production.

After 5 min of incubation at room temperature, both solutions, diluted plasmids and diluted Lipofectamine 2000 reagent, were gently mixed and incubated for 20 min at room temperature. Finally, the plasmid-lipofectamine complexes were added to the cells, and the plate was mixed by gently rocking back and forth. 

After 48 h, the supernatants of transfected cells were collected and filtered through a 0.45-μm PES filter to eliminate detached cells, aliquoted, and subsequently stored at −80 °C until use. Viral titer was determined by qPCR from 100 μL of supernatant.

### 2.4. HaCaT Cell Transduction

HaCaT cells were infected with at least 5 IFU per cell of lentiviral particles containing the *FOXP3Δ2Δ7* ORF. After 72 h of infection, 0.5 μg/mL of puromycin (Cat. No. A1113803, Gibco, Waltham, MA, USA) was added to the cell culture. The cells were left under these conditions for another 72 h. After that, the cells were cultured in media with 0.1 μg/mL of puromycin for two more weeks or until no surviving cells were observed in the noninfected control plate. Following this procedure, two cell lines were obtained: HaCaT-LVX (transduced with the empty vector as control) and HaCaT-FOXP3Δ2Δ7. 

### 2.5. Quantitative PCR

Total RNA was extracted with the Quick-RNA miniprep plus kit (Cat. No. R1058, Zymo Research, Irvine, CA, USA) according to the manufacturers’ instructions. SuperScript II reverse transcriptase (Cat. No. 18064022, Invitrogen, Waltham, MA, USA) and 5 µg of total RNA were used for the cDNA synthesis.

The quantitative PCR (qPCR) was performed on the LightCycler 2.0 equipment (Roche Diagnostics, Basel, Switzerland) with the LightCycler FastStart DNA Master plus SYBR Green I Kit (Cat. No. 03515869001, Roche Diagnostics, Basel, Switzerland). *RPLP0* and *RPL32* were used as reference genes. The sequences of all primers used (*FOXP3*, *SATB1*, *C1R*, *GLI2*, *LAMP3*, *NSG1*, *HSPB8*, *RPLP0,* and *RPL32*) can be found in Table 1.

### 2.6. Proliferation Assay (xCELLigence)

Cells (5000) were seeded in triplicate in an xCELLigence Real-Time Cell Analysis (RTCA) 96-well electronic microplate (E-Plate 96) (Cat. No. 300601010, Agilent Technologies, Inc., Santa Clara, CA, USA) and incubated for 48 h with impedance detection every hour with the xCELLigence RTCA SP station (Agilent Technologies, Inc., Santa Clara, CA, USA). Proliferation was measured as cell index change over time.

### 2.7. Cell Division Tracking

CSFE Cell Division Tracker Kit (Cat. No. 423801, BioLegend, San Diego, CA, USA) was used for cell division analysis. CSFE (carboxyfluorescein succinimidyl ester) reactant was diluted with DMSO (Cat. No. D8418-250ML, Sigma-Aldrich, St. Louis, MO, USA) to prepare a 5 mM solution. In a 60 mm plate, 50,000 cells were seeded and incubated until the next day. A 5 mΜ working solution was prepared in DMEM media without FBS. The cells were incubated with the working solution for an hour, then the media was changed, and fresh DMEM with 10% FBS was added. The cells were kept growing for 72 h until the fluorescence measurement by flow cytometry was performed. The cells were measured right before the incubation for the initial time detection.

### 2.8. Cell Migration Assay

For the cell migration assay, 800,000 cells were seeded in a 6-well plate (Cat. No. 140675, Thermo Fisher Scientific Inc., Waltham, MA, USA) for 24 h until full confluence was reached. A new media with 10 μg/mL mitomycin C (Cat. No. 10107409001, Roche Diagnostics, Basel, Switzerland) was added to avoid proliferation bias. After two hours of incubation with mitomycin C, three straight lines and one horizontal line, as a reference, were manually created with a 200-μL sterile pipette tip on each cell group and its duplicate; the plate was washed twice with PBS to remove nonadherent cells, and finally, fresh media was added.

Photographs were acquired at 0, 12, and 24 h with the 4x objective of a ZEISS inverted microscope (Primo Vert No. 415510-1101-000 model, Carl Zeiss Microscopy GmbH, Jena, Germany), with a ZEISS AxioCam camera (Erc5s model, Carl Zeiss Microscopy GmbH, Jena, Germany), and the ZEN 2012 software (Blue edition, Carl Zeiss Microscopy GmbH, Jena, Germany). Data obtained were analyzed to evaluate the percentage of wound closure with ImageJ software (Image Processing and Analysis in Java; Rasband, W.S., ImageJ, U.S. National Institutes of Health, Bethesda, MD, USA, https://imagej.nih.gov/ij/, accessed on 23 April 2022).

### 2.9. RNAseq

Next-generation RNA sequencing with the Nova Seq 6000 Illumina platform was conducted by Novogen Bioinformatics Technology Co., Ltd. (Beijing, China); independent sample duplicates of total RNA from HaCaT-LVX and HaCaT-FOXP3Δ2Δ7 were sent and preserved in RNAstable (Cat. No. 93221-001, Biomatrica, San Diego, CA, USA).

Retrieved data was analyzed in both open-source platforms, Rstudio (version 1.4.1717) [43], and Galaxy (version 22.05.1) (https://www.usegalaxy.org accessed on 3 February 2022) [44].

The sequencing quality of the returned data (FASTQ archives) was analyzed with FASTQC (version 0.11.9) [45]. Subsequently, alignment to the Homo sapiens genome (version 42) (GRCh38.p13) was performed with the Subjunc aligner (version 2.0.0) [46] from the Rsubread package (version 3.14) [46]. Afterward, BAM files were processed with featureCounts (version 2.0.1) [47]. Normalization of read counts as FPKM (fragments per kilobase of exon per million mapped fragments) was achieved using the DESeq2 tool (version 1.36.0) [48].

Selection of the differentially expressed genes (DEGs) was defined by −1.5 ≤ Log2(fold change) ≥ 1.5 and *p*-value < 0.05 as selection criteria.

### 2.10. Gene Set Enrichment Analysis

From the data obtained by the DESeq2 tool, two matrices were generated, the first in a tab-delimited text file with the normalized expression data in FPKM and the second in a cls file with the identification of the phenotypes. Both matrices were loaded in GSEA software (version 4.2.2) [49], the “Hallmark” gene set database (h.all.v2022.1.Hs.symbols.gmt) was selected, and for the *Chip Platform*, the “Human_Gene_Symbol_with_Remapping_MsigDB.v7.5.1.chip” option was chosen. One thousand phenotype permutations were established, and the Signal2Noise parameter was selected in the *metric for ranking genes* configuration. Finally, an FDR value < 0.25 was chosen to identify significant enriched pathways.

### 2.11. Statistics

All experiments were performed with at least two independent replicates; the two-way analysis of variance (ANOVA) and unpaired Student *t* tests were used to calculate differences between groups. A *p*-value < 0.05 was considered statistically significant.

## 3. Results

### 3.1. Augmented FOXP3 Expression in Cervical Cancer

To identify the expression profile of *FOXP3* in CC samples, an expression analysis in OncoDB (https://oncodb.org/ accessed on 7 September 2022) [50] was performed. Results from CC samples (*n* = 304) compared against normal tissue (*n* = 22) showed a significant increase in *FOXP3* mRNA expression in tumor samples (Figure 1a). To identify if *FOXP3* expression was related to the three most common CC subtypes, data collected from “Cervical Squamous Cell Carcinoma and Endocervical Adenocarcinoma (TCGA, Firehoses Legacy)” (*n* = 310) in cBioPortal was assessed (https://www.cbioportal.org/ accessed on 7 September 2022) [51,52]. The findings depict the *FOXP3* mRNA expression level by CC subtype as well as the frequency of positive samples for *FOXP3*; deriving from those values, *FOXP3* was detected in 98.8% of samples with SCC, 100% of ADC samples, and 80% of ADSC; moreover, *FOXP3* expression average in SCC was higher compared with the other two subtypes (Figure 1b). 

### 3.2. FOXP3Δ2 Is the Most Prevalent Variant in Cervical Cancer Samples, and Its Expression Is Highly Differential in Metastatic Stages

To deepen into the *FOXP3* mRNAs isoforms expressed in cervical cancer, data from TCGA-CESC was analyzed in ISOexpresso (http://wiki.tgilab.org/ISOexpresso/ accessed on 10 October 2022) [53] and TSVdb (http://tsvdb.com/ accessed on 10 October 2022) [54] web-based platforms. Results obtained from the analyses showed that tumor samples express three variants with a differential frequency, *FOXP3Δ2* with a frequency of 74%, *FOXP3Δ7* with 17%, and *FOXP3 X1* with 9% (Figure 2a). TSVdb results showed that the most frequent variant, *FOXP3Δ2,* was expressed with a high difference between normal and metastatic tissue (Figure 2b). The *FOXP3Δ7* variant was observed with slightly increased expression in primary solid tumors, but no difference was observed for normal and metastatic samples (Figure 2c). Unfortunately, the *FOXP3Δ2Δ7* variant is not included in these databases. Therefore, no information related to this isoform was available in either database.

### 3.3. HPV16 Infection Could Increase FOXP3 Expression Levels in Cervical Cancer

The main etiological factor for developing CC is persistent infection with high-risk HPV, such as 16 or 18 genotypes. To identify if there is an association between the HPV presence and the *FOXP3* increase, *FOXP3* expression levels in CC samples HPV16+ and HPV18+ from OncoDB were analyzed. Results obtained from OncoDB showed an increase in *FOXP3* expression in the presence of both genotypes but with a marked difference in HPV16+ samples, in which a greater tendency of overexpression was observed compared to the HPV18+ and HPV(-) CC samples (Figure 3a).

After that, it was of interest to determine the status of *FOXP3* expression in CC-derived cell lines HPV16+ (SiHa and Ca Ski), HPV18+ (HeLa and SW756), HPV(-) (C33A), and nontumorigenic keratinocytes (HaCaT). The relative expression level of *FOXP3* was assessed by qPCR and compared to the expression of HaCaT cells. The results indicated high relative expression in SiHa cells. However, CaSki cells, equally positive for HPV16, showed decreased expression of *FOXP3*. On the other hand, HPV18-positive cell lines express differentially *FOXP3* levels, HeLa cells with a subexpression and a slight increase in the SW756 line. Regarding the C33A cell line (HPV negative), a trend of overexpression was observed (Figure 3b).

The evidence so far supports that *FOXP3* expression tends to increase in cervical cancer and HPV-positive cervical samples. However, in the panel of CC-derived cell lines analyzed, only the SiHa and C33A cell lines seem to maintain this feature. To have some insights into the molecular implications of FOXP3 expression, we selected the SiHa cell line. Therefore, the FOXP3 ORF from this cell line was isolated, cloned, and sequenced. The findings showed that SiHa cells express the *FOXP3Δ2Δ7* variant (Figure 4b), an isoform of FOXP3 are poorly studied; then, the derived question was to know whether the *FOXP3Δ2Δ7* varies the shapes and behavior of nontumorigenic keratinocytes.

### 3.4. Exogenous Expression of FOXP3Δ2Δ7 Promotes Cell Proliferation, Division, and Migration

Once the *FOXP3Δ2Δ7* ORF from SiHa was subcloned and stably transduced into HaCaT cells, validation of the lentiviral transduction was assessed by qPCR and RNAseq. The results demonstrated a significant increase in *FOXP3’s* relative expression in HaCaT-FOXP3Δ2Δ7 (Figure 4a). Furthermore, the RNAseq analysis of the transduced model shows abundant reads covering the genomic region corresponding to the *FOXP3* gene. It was also possible to observe the lack of exons 2 and 7, confirming that the mRNA isoform expressed by HaCaT-transduced cells was the *FOXP3Δ2Δ7* version (Figure 4b).

To identify the effect of *FOXP3Δ2Δ7* exogenous expression in cell proliferation, cell index determination by impedance with the xCELLigence RTCA platform was performed (Figure 5a); results exhibit an increase in cell index when compared with control. Additionally, cell division tracking assays revealed a diminished fluorescence in *FOXP3Δ2Δ7*-expressing cells, which indicates a higher cell division rate (Figure 5b). Further, wound healing assays showed that *FOXP3Δ2Δ7* significantly promoted the migration process (Figure 5c).

### 3.5. Exogenous Expression of FOXP3Δ2Δ7 Induces the Transcription of Protumoral Genes and the Enrichment of Immunological and Oncogenic Pathways

Next-generation sequencing of total mRNA was performed to comprehend the molecular mechanisms behind the protumorigenic activities induced by the exogenous expression of *FOXP3Δ2Δ7*. RNAseq assays highlighted differentially expressed genes. Using the selection criteria of −1.5 ≤ Log2 (fold change) ≥ 1.5 and a *p*-value < 0.05, the results showed 23 upmodulated genes and 27 downmodulated genes. As expected, *FOXP3* was the most overexpressed gene with a Log2(fold change) of 6.84, and among the 23 upregulated genes, numerous are associated with protumorigenic functions (Figure 6a). To validate the RNAseq results, the genes *SATB1*, *C1R*, *GLI2*, *LAMP3*, *NSG1,* and *HSPB8* were selected and analyzed by qPCR (Figure 6b,c); the results show a global trend towards the upregulation of pro-oncogenic genes.

Furthermore, for the recognition of *FOXP3Δ2Δ7* modulated pathways, enrichment analysis was performed by using the GSEA software. For a confident selection of modulated pathways, an FDR q-value < 0.25 was used. Among the pathways modulated by *FOXP3Δ2Δ7*, several are involved in the immune response (e.g., complement, inflammatory response, IL2/STAT5 signaling, IL6/JAK-STAT signaling), and others have been associated with oncogenic processes (e.g., KRAS, WNT/beta-catenin, hypoxia) (Figure 7).

## 4. Discussion

FOXP3, a member of the transcriptional factor family forkhead/winged-helix, is commonly known as a master regulator of regulatory T CD4+ CD25+ cells, which promotes homeostasis and immunological tolerance. These functions are established through the transcriptional activation or repression of approximately 700 genes and miRNAs related to the TCR pathway, cell communication, and transcriptional regulation [55].

Nowadays, it is known that FOXP3 expression occurs not only in hematopoietic cells but also in cancer cells. Karanikas et al. reported high expression levels of FOXP3 in 25 cancer-derived cell lines [26]. Several studies have reported the FOXP3 expression increase in tumor cells, and in some cases with tumor suppressor activity, as in ovarian, prostate, and breast cancer [29,32,56], but also as an oncoprotein, as in gastric, bladder, lung, and cervical cancer, among others [37,38,40,57].

Since 2012, when the first report about FOXP3’s increase in CC was published [39], its biological role in the development of cervical pathology has been unclear. Initial findings showed that FOXP3 overexpression was especially marked in CC samples [22,39,40]; in this work, such increased expression was also evident in HPV+ CC samples, particularly within the most frequent genotype, HPV16. These findings suggest that the increase in FOXP3 does not occur solely due to the presence of HPV but could also be dependent on the infecting genotype; Zeng et al., in 2012, established the hypothesis that the virus uses FOXP3 increase as an essential mechanism to control the immune system and to preserve the active infection [39]. To confirm or deny this last postulate, it is necessary to carry out more elaborated approaches.

On the other hand, the analysis of FOXP3 variants’ frequencies highlights the FOXP3Δ2 isoform as the most common in CC samples. To the best of our knowledge, this is the first work analyzing FOXP3 variants in cervical cancer, particularly the *FOXP3Δ2Δ7* variant, which was identified in the SiHa cell line. The relevant feature of losing the coding exon two region, or Δ2, is the absence of a repression domain, which is believed to have tumor suppressor functions due to a possible transcriptional restriction of oncogenes [58]. Therefore, isoforms lacking could allow free transcription of normally repressed genes. On the other hand, the full-length variant, in which the coding exon two is present, should have opposite activities, and this is confirmed by the observations using breast and ovarian cancer cells, in which the introduction of the FOXP3-FL variant decreases tumor growth, inhibits proliferation, and decreases migration and invasion [29,32]. Additionally, the coding exon two loss was associated with greater aggressiveness and chemo-resistance in bladder cancer [57]. Moreover, in nonsmall cell lung cancer, it was related to increased proliferation, migration, and invasion [38]. The loss of the coding exon 7 (Δ7) was previously documented in an isoform of FOXP3 [15], FOXP3Δ7 lacks an 81-bp region that contains part of the leucine zipper domain of the protein, which is necessary for protein-protein interactions, and it was postulated that mutations in the exon 7 prevent protein dimerization and, therefore, DNA binding [59]. Mutations in exon 7 were associated with an autoimmune disease called IPEX [60,61]. Nevertheless, Mailer et al., working with the FOXP3Δ2Δ7, found that the lack of exon 7 does not affect protein dimerization or interaction with RUNX1, NFAT, or NF-kB [62]. However, the exogenous expression of FOXP3Δ2Δ7 failed to induce the typical Treg-associated phenotype in lymphocytes [62]. The latter indicates that FOXP3Δ2Δ7 has a different function compared with FOXP3-FL. In this work, we report the expression of FOXP3Δ2Δ7 in SiHa cells. The lentiviral transduction of FOXP3Δ2Δ7 in nontumorigenic keratinocytes stimulates cell proliferation and division. These results agree with the observations of Luo et al. [40]. They induced the silencing of FOXP3 by employing RNA interference in SiHa cells and observed a decrease in proliferation, migration, and invasion [40]. Evidence of an FOXP3 oncogenic behavior was also observed in gastric cancer [37]. Similarly to our work, the exogenous expression of FOXP3 induces both proliferation and migration, however, the authors do not provide information about the FOXP3 isoform [37]. All this evidence indicates the importance of determining not only the expression of FOXP3 but also the isoform that the cell is expressing.

Functional analyzes establish clear evidence of the protumoral capacity of *FOXP3Δ2Δ7* in vitro, to delve into the molecular mechanisms underlying the biological activities of this isoform transcriptomic analysis were performed. Among the overexpressed genes, *SATB1* highly increased. This gene is involved in the epithelial-mesenchymal transition [63,64,65,66], and its overexpression has been associated with a worse prognosis in solid tumors [67]. In CC, it was identified as highly expressed in those women who presented advanced stages of the disease and through a Kaplan–Meier analysis, it was possible to associate it as a worse prognosis marker [68].

Other gene upregulated was *GLI2*, it has been identified as a potent oncogene that mediates the hedgehog signaling pathway [69,70,71]; in CC it was shown that its subexpression inhibited the growth and migration of cell lines derived from cancer, and it was also associated as a poor prognostic marker based on TCGA data [72].

Regarding C1R, there is no information related to CC; however, what has been published to date relates this gene to tumor growth, vascularization, and invasion in cutaneous squamous cell carcinoma [73,74]. High levels of C1R were reported in nonsmall cell lung cancer compared to control groups. Interestingly, the high levels of this gene were associated with an increased risk of death [75].

LAMP3 was associated with increased migration in an overexpression model derived from a cervical cancer cell line, and an in vivo metastasis assay demonstrated that 9 of 11 LAMP3 overexpressing mouse models elicited a process of distal metastasis with invasion of the lymphovascular space. Additionally, they demonstrated that in 100% of the CC samples, the gene was found to be overexpressed [76]. 

The enrichment analysis highlights important pathways modulated by *FOXP3Δ2Δ7* expression, some of them closely related to the immune response. It has been reported that FOXP3 inhibits the activity of NFAT and NF-κB, two important transcription factors essential for cytokine expression [77]. These results agree with those reported in melanoma, which indicates that FOXP3 in cancer cells modulates the expression of molecules associated with immunity, in addition to contributing to the recruitment of Treg lymphocytes, which establishes a suppressive activity that favors tumor progression [78].

On the other hand, several of the signaling pathways identified as enriched are recognized for their oncogenic potential. The KRAS signaling pathway is highly involved in cell proliferation and division; it belongs to the RAS/MAPK pathway, and in the oncogenic context, KRAS has very important effects in the carcinogenesis process of endometrial, colorectal, bladder, breast, and cervical cancer [79,80,81,82,83,84,85,86].

Regarding the WNT/β-catenin signaling pathway, it is involved in both the proliferation process and migration; in CC, it is recognized as a practically necessary pathway for cell transformation, which leads to an increase in the aforementioned processes [87], additionally, it is interesting to mention that the *SATB1* gene is related to the process of migration, and with the WNT/β-catenin pathway, this has been reported in colorectal cancer by promoting tumorigenesis, progression, and epithelial-mesenchymal transition [63,88].

Ultimately, it is important to mention that *FOXP3Δ2Δ7* modulates the hypoxia signaling pathway, this pathway involves genes activated in response to low oxygen concentrations. In cancer, this phenomenon is highly relevant for the establishment of cancer hallmarks and the activation of cellular processes that include proliferation, survival, tumor invasion, and metastasis [89,90,91]. In CC, as expected, high degrees of hypoxia are associated with a poor prognosis [92] and, interestingly, also with resistance to chemotherapeutics [93,94]. As well, it is pertinent to acknowledge the role of *LAMP3* in the pathway, as Mujcic et al. demonstrated in a cohort of human cervix tumors an increased expression of the gene as a consequence of hypoxia but furthermore, they found that LAMP3 is an essential regulator of hypoxia-driven nodal metastasis [95].

Future research should aim to evaluate the role of *FOXP3Δ2Δ7* on the immune response regulation in the immune cells and tumor cells at the tumoral niche in CC.

## 5. Conclusions

The behavior of the FOXP3 expression during cervical cancer development was observed to increase. FOXP3 expression was also increased in HPV-positive samples, especially in those positive for HPV16. An analysis of the isoforms expressed in this pathology indicates the importance of the loss of the exon coding two of FOXP3. Functional studies suggest that isoform *FOXP3Δ2Δ7* presents a protumorigenic behavior when introduced into keratinocytes since it induces cell proliferation, migration, and division. Additionally, the transcriptomic analysis highlights the importance of this isoform in the modulation of pathways, such as interferon, apoptosis, TNF, hypoxia, IL-2, IL-6, and the inflammatory response, among others. The latter reveals the critical role the FOXP3 isoforms could play in modulating the immune response. Subsequent studies evaluating FOXP3 isoforms and their effect on immune cells will provide a better understanding of the immunomodulatory effects of FOXP3 expression in tumor cells.

## Figures and Tables

**Figure 1 cancers-15-00347-f001:**
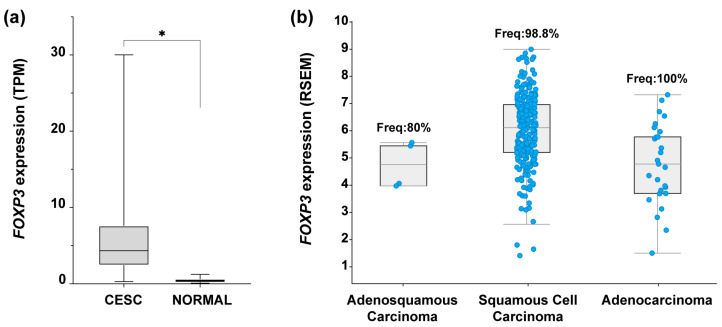
*FOXP3* expression levels in cervical cancer. (**a**) Comparison of *FOXP3* expression levels in transcript per million (TPM) from cervical epithelial squamous carcinoma (CESC) samples (*n* = 304, avg = 5.7, median = 4.3) vs. normal tissue (NORMAL) (*n* = 22, avg = 0.4, median = 1), Log2(Fold Change) = 2.1; (**b**) *FOXP3* expression abundance in RNA Seq by expectation-maximization (RSEM) from TCGA-CESC data; RSEM values are expressed as RNA Seq V2 RSEM (Log2(value + 1)). Each blue dot represents a patient sample positive for *FOXP3* expression: adenosquamous carcinoma, *n* = 4 of 5 (frequency = 80%), squamous cell carcinoma, *n* = 253 of 256 (frequency = 98.8%), adenocarcinoma, *n* = 27 of 27 (frequency = 100%); TCGA-CESC, *n* = 308 patients. (*) depicts statistical significance with a *p*-value < 0.05.

**Figure 2 cancers-15-00347-f002:**
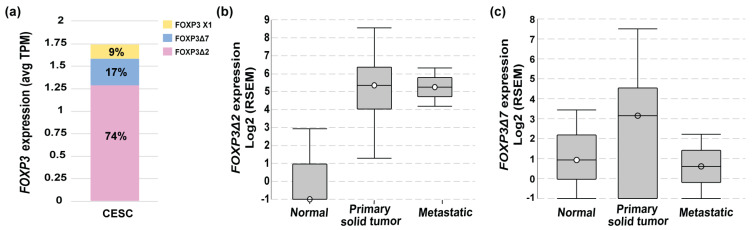
*FOXP3* variant levels in cervical cancer. (**a**) *FOXP3* expression levels and variant frequencies in transcripts per million (TPM) from TCGA-CESC data (*n* = 305); (**b**) *FOXP3Δ2* expression abundance in RNA Seq by expectation-maximization (RSEM) in normal (*n* = 1 of 3), primary solid tumor (*n* = 273 of 304) and metastatic (*n* = 2 of 2); TCGA-CESC, *n* = 309; (**c**) *FOXP3Δ7* expression abundance in RNA Seq by expectation-maximization (RSEM) in normal (*n* = 2 of 3), primary (*n* = 222 of 304), and metastatic solid tumors (*n* = 1 of 2); TCGA-CESC *n* = 309.

**Figure 3 cancers-15-00347-f003:**
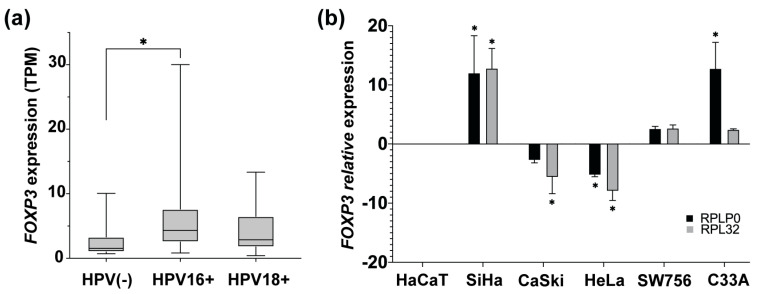
*FOXP3* expression levels in cervical cancer samples and cervical cancer-derived cell lines. (**a**) Comparison of *FOXP3* expression levels in transcript per million (TPM), from CC samples HPV16+ and HPV18+ vs. HPV(-) CC samples; (**b**) detection of *FOXP3* relative expression levels by qPCR in CC-derived cell lines HPV16+ (SiHa and CaSki), HPV18+ (HeLa and SW756), and HPV(-) (C33A); *RPLP0* and *RPL32* were used as reference genes. The expression of FOXP3 in HaCaT cells was used as a calibrator. (*) depicts statistical significance with a *p*-value < 0.05.

**Figure 4 cancers-15-00347-f004:**
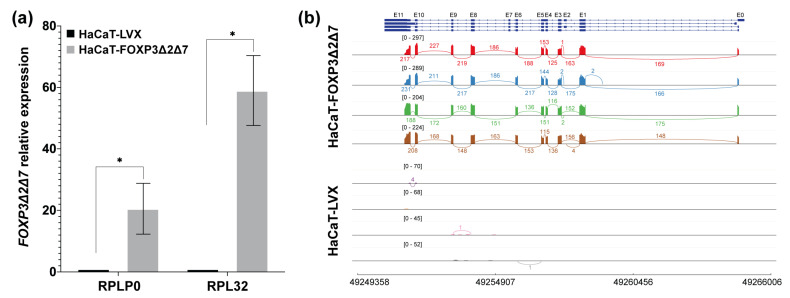
Validation of the establishment of the *FOXP3Δ2Δ7* exogenous expression model. (**a**) *FOXP3Δ2Δ7* relative expression levels were detected in the cell model by qPCR; *RPLP0* and *RPL32* were used as reference genes. The expression of *FOXP3* in HaCaT-LVX was used as a calibrator to calculate the relative expression; (**b**) Sashimi plot of the RNAseq assays from replicates of HaCaT-FOXP3Δ2Δ7 and HaCaT-LVX cell lines, reads covering the *FOXP3* genomic region are shown as colored peaks, and each numbered curved line represents an interexonic junction with the detection level. Upper blue boxes and connecting blue lines represent exonic and intronic regions. (*) depicts statistical significance with a *p*-value < 0.05.

**Figure 5 cancers-15-00347-f005:**
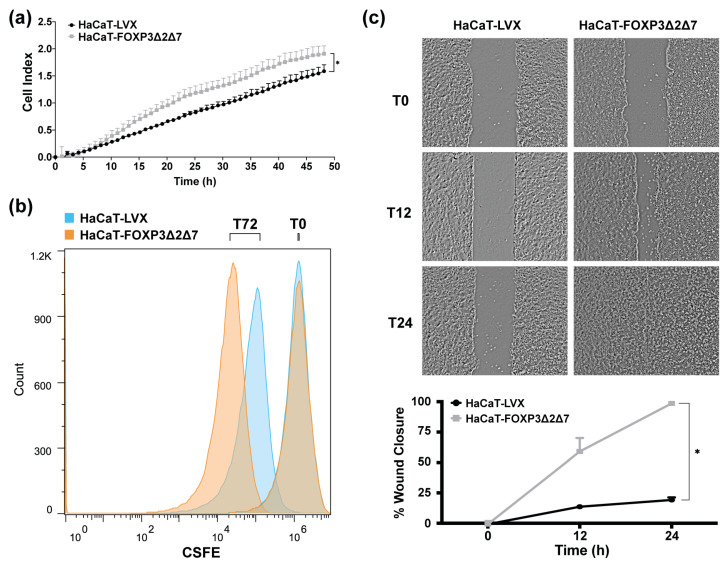
Evaluation of biological effects by *FOXP3Δ2Δ7* exogenous expression in vitro. (**a**) Real-time cell proliferation analysis with xCELLigence RTCA platform for 48 h; values are expressed as cell index; (**b**) cell division tracking with CSFE reagent, fluorescence measurement was achieved with flow cytometry at the initial time point and 72 h later; (**c**) depictive images of wound healing assay and plot of wound closure percentage during 24 h; photographs were acquired at 4× amplification and processed with ImageJ software. (*) depicts statistical significance with a *p*-value < 0.05.

**Figure 6 cancers-15-00347-f006:**
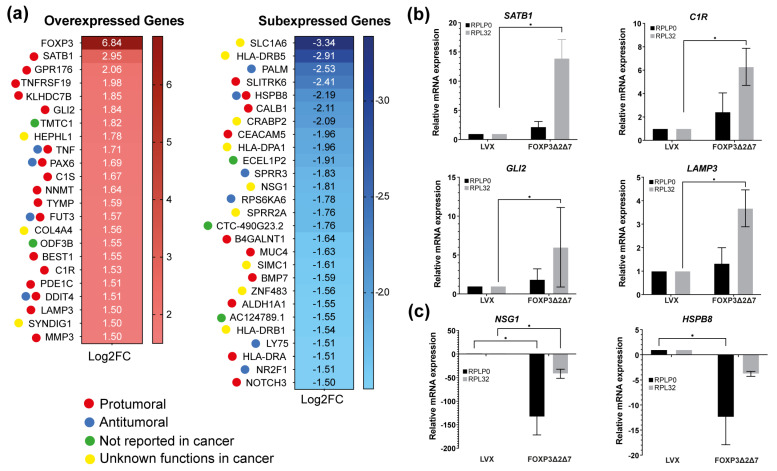
Identification of differentially expressed genes by *FOXP3Δ2Δ7* exogenous expression. (**a**) Differentially expressed genes heatmap selected by −1.5 ≤ Log2(fold change) ≥ 1.5, *p*-value < 0.05; code color indicates a review of what has been previously published about that gene; (**b**) relative mRNA expression of upregulated genes by qPCR; *RPLP0* and *RPL32* were used as reference genes; (**c**) relative mRNA expression of downregulated genes by qPCR; *RPLP0* and *RPL32* were used as reference genes. (*) depicts statistical significance with a *p*-value < 0.05.

**Figure 7 cancers-15-00347-f007:**
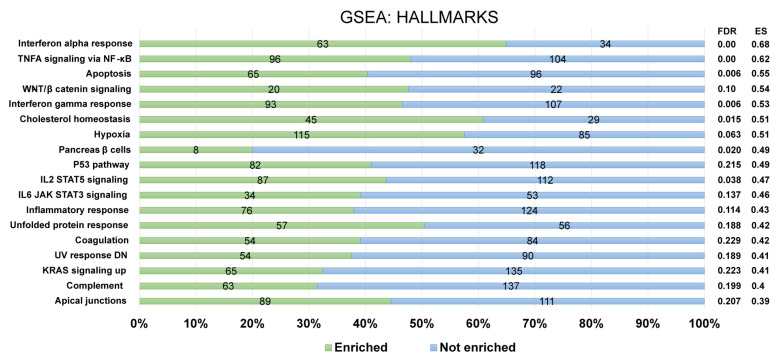
Identification of enriched biological pathways modulated by *FOXP3Δ2Δ7* exogenous expression. Enrichment analysis from the Hallmark gene set collection was assessed by GSEA software. The left panel depicts the standard names of the statistically significant pathways, and the right panel shows the false discovery rate (FDR) and enrichment score (ES) values. The numerals at the bar plot indicate the number of enriched (green) and not enriched (blue) genes within each pathway. FDR < 0.25 was established as a selection criterion. A percentage scale is shown at the bottom for better visualization.

**Table 1 cancers-15-00347-t001:** Primers sequences.

Gene	Forward	Reverse	Amplicon Size
*FOXP3* ^1^	5′ ACA AGC CAG GCT GAT CCT T 3′	5′ CAC ATC CAG GGC CTA TCA TC 3′	1347 bp
*FOXP3* ^2^	5′ CAA GTT CCA CAA CAT GCG ACC 3′	5′ GCT CTC CAC CCG CAC AAA 3′	208 bp
*SATB1* ^2^	5′ CCT CAG CCA GAA CGT GAT GC 3′	5′ GAC TCT GCT GGA GAG GCC A 3′	236 bp
*C1R* ^2^	5′ AAG ATT CCT CGG TGC TTG CC 3′	5′ GTT GCT TTG CGC TTC GTG TT 3′	216 bp
*GLI2* ^2^	5′ CAA CAA TGA CAG TGG CGT GG 3′	5′ CTG CCA CTG AAG TTT TCC AGG 3′	297 bp
*LAMP3* ^2^	5′ ACA TGC GGT GGT GAT GTT CC 3′	5′ AGG CAG AGA CCA ACC ACG AT 3′	219 bp
*NSG1* ^2^	5′ TTC CTC ACC TGC GTC GTC TT 3′	5′ AAC TTG CCC ATC CCG CTA AG 3′	297 bp
*HSPB8* ^2^	5′ GGT GGC ATT GTT TCT AAG A 3′	5′ TAC TGG CAT CTC AGG TAC AG 3′	208 bp
*RPLP0* ^2^	5′ CCT CAT ATC CGG GGG AAT GTG 3′	5′ GCA GCA GCT GGC ACC TTA TTG 3′	95 pb
*RPL32* ^2^	5′ GCA TTG ACA ACA GGG TTC GTA G 3′	5′ ATT TAA ACA GAA AAC GTG CAC A 3′	320 pb

^1^ Primer used for gene cloning. ^2^ Primer used for quantitative PCR.

## Data Availability

The RNAseq data presented in this study are openly available in the Gene Expression Omnibus (GEO) database repository (https://www.ncbi.nlm.nih.gov/geo, accessed on 12 December 2022), with the GEO accession number GSE221082.

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
