# Peer review of "FOXP3 Isoforms Expression in Cervical Cancer: Evidence about the Cancer-Related Properties of FOXP3Δ2Δ7 in Keratinocytes"

_cancers, 2023, doi:10.3390/cancers15020347_

Round 1

Reviewer 1 Report

Dear authors the work is well done and nice.

I have only one suggestion that should be correct; remove the word "were gifted from Didier Trono" and put this information in the acknowledgment, please.

Author Response

Thank you very much for your comments and your opinion about our manuscript.

We have made the changes you mention.

Thanks again.

Reviewer 2 Report

This article presents the role of FOXP3Δ2Δ7 in cervical cancerogenesis with several different methods including RNA-seq that provide multi-facet information about this protein production either in vitro or in vivo. Although the key factor in cervical squamous cancer is well known the prediction of progression, target therapy response and aggressiveness has yet to be investigated. The authors highlight that high expression levels of FOXP3 were demonstrated in in 25 cancer-derived cell lines. In addition, they mention several studies that reported the FOXP3 expression increase in ovarian, prostate, breast, gastric, bladder, lung and, cervical cancer. In the present study the authors make efforts to clarify the role of different isoforms of FOXP3 in cervical cancer pathogenesis because they suppose that they can have different impact on tumor vascularization, growth and metastasis. Finally, they concentrate on FOXP3Δ2Δ7 as the most impactful protein demonstrating its role in cell proliferation and cell index determination with different methods. The authors convincingly demonstrated the interplay between HPV and FOXP3, however it would be better to compare these results with HPV-independent squamous carcinomas (although they are rather rare).

I suppose that the article should be more traditionally written because the results are too flourish teeming with discursive moments.

In addition, it should be mentioned that the list of the references includes 50% papers older than 5 years and should be refreshed a little bit.

After these corrections the article could be recommended for the publication in Cancers.

Author Response

Thanks for your comments and suggestions about our manuscript.

…The authors convincingly demonstrated the interplay between HPV and FOXP3, however it would be better to compare these results with HPV-independent squamous carcinomas (although they are rather rare).

It would be very interesting to analyze HPV-negative samples. However, this is difficult since almost 99% of cervical cancer is related to HPV infection. Our work observed that the C33 cell line, which is HPV-negative, also shows elevated FOXP3 expression. Therefore, it could be speculated that FOXP3 induction could be a common mechanism in HPV+ and HPV- cervical cancer. More work is needed in this regard

I suppose that the article should be more traditionally written because the results are too flourish teeming with discursive moments.

Thanks for the suggestion; we have revised the manuscript and rewritten it to be more concise and direct.

In addition, it should be mentioned that the list of the references includes 50% papers older than 5 years and should be refreshed a little bit.

More current references were added to the manuscript. Thanks for the comment.

After these corrections the article could be recommended for the publication in Cancers.

We are grateful for the suggestions that have improved our manuscript.